# On the Origin and Fate of Reactive Oxygen Species in Plant Cell Compartments

**DOI:** 10.3390/antiox8040105

**Published:** 2019-04-17

**Authors:** Martina Janků, Lenka Luhová, Marek Petřivalský

**Affiliations:** Department of Biochemistry, Palacký University in Olomouc, Šlechtitelů 27, CZ-78371 Olomouc, Czech Republic; janku.martina@seznam.cz (M.J.); lenka.luhova@upol.cz (L.L.)

**Keywords:** cell wall, chloroplasts, cytoplasmic membrane, cytosol, glyoxysomes, mitochondria, peroxisomes, plant cell, reactive oxygen species

## Abstract

Reactive oxygen species (ROS) have been recognized as important signaling compounds of major importance in a number of developmental and physiological processes in plants. The existence of cellular compartments enables efficient redox compartmentalization and ensures proper functioning of ROS-dependent signaling pathways. Similar to other organisms, the production of individual ROS in plant cells is highly localized and regulated by compartment-specific enzyme pathways on transcriptional and post-translational level. ROS metabolism and signaling in specific compartments are greatly affected by their chemical interactions with other reactive radical species, ROS scavengers and antioxidant enzymes. A dysregulation of the redox status, as a consequence of induced ROS generation or decreased capacity of their removal, occurs in plants exposed to diverse stress conditions. During stress condition, strong induction of ROS-generating systems or attenuated ROS scavenging can lead to oxidative or nitrosative stress conditions, associated with potential damaging modifications of cell biomolecules. Here, we present an overview of compartment-specific pathways of ROS production and degradation and mechanisms of ROS homeostasis control within plant cell compartments.

## 1. Introduction

Subcellular compartmentalization in eukaryotic cells forms the basis for highly selective separation of biochemical reactions and metabolic pathways, and delimit their potential mutual interferences [1]. Current knowledge indicates that within each compartment of eukaryotic cells, specific redox characteristics have evolved [2,3]. Unique redox aspects of different cell compartments are integrated at the molecular level by a group of small diffusible and reactive molecules, reactive oxygen species (ROS), which provide communication between intracellular compartments [4]. ROS are produced in all forms of aerobic life by divergent reactions including a partial reduction of the molecular oxygen [5,6]. Previously, ROS were described as toxic by-products of the aerobic metabolism and their increased levels were connected with multiple forms of cellular damage mediated by oxidative modifications of cell biomolecules. Nowadays, the accumulated evidence suggest that ROS function as key redox signaling and effector molecules in vital biological processes such as cell growth, differentiation, proliferation and responses to a wide spectrum of external stimuli [7,8]. In plant cells, signaling networks of ROS are closely connected to central physiological processes of energy generation and consumption such as respiration, photosynthesis and photorespiration and in plant responses to abiotic and biotic stress conditions [9,10,11].

In general, signaling functions of ROS are related to tightly regulated site- and time-specific modulations of ROS levels, whereas uncontrolled ROS accumulation as a consequence of either induced production or defective catabolism, or their combination, is associated with damaging oxidative effect leading to cellular damages or even to the cell death [12]. ROS can modify proteins structures and functions through oxidative reactions namely with protein thiol groups or iron-containing clusters. On the other hand, toxic effects of accumulated ROS are also exploited as chemical defense within immune repertoires of diverse organisms against invading pathogens, such as the well-described ROS burst leading to the plant cell death within the plant hypersensitive response to biotrophic pathogens [13].

Individual ROS show a high variability in their chemical properties, reactivity and involvement in redox signaling pathways [14]. As a general rule, radical forms of ROS are more reactive compared to non-radical ROS. Moreover, ROS reactivity is significantly affected by presence of free Fe^2+^ ions involved in formation of highly reactive hydroxyl radical in a pathway termed the Fenton reaction, which results in production of hydroxyl radicals as the most reactive ROS form with a single unpaired electron, capable to react with virtually all biomolecules. High reactivity of hydroxyl radicals leads to subsequent cellular damages, including changes in protein structures, lipid peroxidation and membrane destruction [15]. However, recent findings support the suggested role of the hydroxyl radical as being more than a destructive agent, as its oxidative properties can facilitate seed germination, growth, stomatal closure, reproduction, immune responses, plant cell death and adaptations to stress conditions [16].

Singlet oxygen (^1^O_2_) is formed in photosynthetic membranes in reactions between triplet-state chlorophyll molecules and the molecular oxygen [17]. Observed half-life of ^1^O_2_ is approximately 3 μs and it has been shown that a minor portion of singlet oxygen is able to diffuse to a distance of several nanometers, where it can react with diverse biomolecules and potentially mediate its signaling functions [18,19]. Enzyme activity of lipoxygenase represents an additional source of ^1^O_2_ in plants [20]. Highly reactive ^1^O_2_ can be efficiently removed by quenching with low-molecular lipophilic compounds like β-carotene and tocopherol, or by a scavenging action of protein D1 in photosystem II [21]. 

Superoxide anion radical (O_2_^−^) is a highly reactive nucleophilic species with a half-life about 1 μs, which often serves as an initiator of reaction cascades generating other ROS, namely hydrogen peroxide. Superoxide occurs with a similar half-life as singlet oxygen, but strongly differs in its target reaction partners. Polyunsaturated fatty acid within thylakoid membranes are considered the main reaction partners of ^1^O_2_, whereas superoxide preferentially reacts with other radical compound including nitric oxide and components of protein hem-containing and non-hem iron centers [22].

Hydrogen peroxide (H_2_O_2_) is the least reactive non-radical ROS. Its signaling pathways are mediated mainly through interactions with proteins containing redox-sensitive moieties, such as metal centers or cysteine thiols, whose oxidation controls protein biological activity [23,24]. H_2_O_2_ can diffuse across lipid membranes through the aquaporin channels, which is a crucial feature for H_2_O_2_ intracellular signaling [25]. In plants, it was demonstrated that H_2_O_2_ transport could also be mediated by aquaporin homologues from the plasma membrane intrinsic factor (PIP) family and by the tonoplast intrinsic protein (TIP) [26]. Excess H_2_O_2_ is known to trigger chloroplast and peroxisome autophagy and programmed cell death in plants [27].

## 2. Subcellular Localization and Functions of ROS Production in Plant Cells

On a quantitative scale, in contrast to animals where mitochondria play a predominant role in the ROS biogenesis in most cell types, plant cells generate ROS in a highly variable manner by multiple pathways depending on the plant tissue, developmental stage and external conditions [9,28]. ROS metabolism has been extensively studied in all plant cell compartments including cell wall, apoplast, plasma membrane, cytosol, mitochondria, chloroplasts, peroxisomes and glyoxysomes [29]. More stable ROS species like hydrogen peroxide can even diffuse, cross cell membranes and transport their signaling effects into other compartments (Figure 1). Moreover, ROS cooperate and cross-talk with signaling pathways of plant hormones such as abscisic, jasmonic and salicylic acid or ethylene [30,31].

### 2.1. Cell Wall, Apoplast and Cytoplasmic Membrane

The cell wall, apoplast and cytoplasmic membrane form crucial boundary compartments of plant cells securing their integrity vital for intracellular processes and communication with external environment including interactions with microbial symbionts or pathogens [32,33]. An important portion of ROS produced under physiological conditions and in plant responses to stress stimuli is formed within these compartments, which represent the major site of ROS production in plant-pathogen interactions [34]. Intensive ROS production termed as the oxidative burst is activated upon the recognition of pathogens or other biotic stress stimuli by immune receptors localized in plasma membrane, which triggers signaling pathways leading to transient localized increase of ROS [35]. In plants, receptor-like kinases (RLKs) represent key elements of the communication between the external environment and the cellular interior. Apoplastic ROS production occurs often following activation of RLK signaling in a wide array of cellular processes [36]. Thus, intricate interconnections exist between RLKs, extracellular ROS generation and ROS signal transduction and perception within the plant ROS sensing machinery. ROS can further diffuse as H_2_O_2_ inside the plant cells via membrane aquaporin channels [25]. Furthermore, apoplastic ROS are key players in physiological processes such as cell growth, which is particularly dependent on ROS involvement in increasing extensibility of the cell wall during root hair and pollen tube elongation, as well as in leaf growth [37,38]. It is known that hydroxyl radical promotes cell elongation by loosening the cell wall by oxidative cleavage of pectins and xyloglucans. On the other hand, H_2_O_2_ can initiate polysaccharide cross-linking and protein disulfide bond formation which restricts elongation growth. Several apoplastic and plasma-membrane proteins are involved in the regulation of apoplastic balance between hydroxyl radical and H_2_O_2_, which regulates cell expansion by direct interactions with cell wall components and also ROS interactions with intracellular transcriptional networks and cytoskeleton [37].

Plant-specific class III peroxidases, as members of a large multigene family of peroxidases (POXs, EC 1.11.1.7) are localized in the plant cell wall and comprise a significant source of apoplastic ROS [39]. Peroxidases contain eight conserved cysteine residues, one or more glycosylation sites, and a signal peptide which divert them into cell wall, apoplast or vacuole [40]. Peroxidases show a broad substrate specificity and can operate either in ROS-consuming or ROS-generating mode through reaction mechanisms based on peroxidation or hydroxylation cycle, respectively [41,42,43,44]. In ROS-consuming peroxidase mode, one oxygen atom from H_2_O_2_ is transferred to the hem group of the ferric enzyme in the ground state and the first intermediate is formed. Then an electron from a reducing substrate generates the second intermediate and the cycle is finished by regeneration of ferric enzyme [41]. H_2_O_2_ reduction catalyzed by peroxidases occurs with simultaneous oxidation of electron donors such as phenolic compounds [45]. Yet, the presence of certain reductants is not the only condition for peroxidases functions; the second one is the apoplast alkalinization. In ROS-producing mode, peroxidases produce ROS due to the oxidase cycle forming the third intermediate. Plant class III peroxidases are responsible for production of apoplastic ROS and involved in stress signaling after pathogen attack, wounding and under abiotic stress stimuli [46]. Beside their function in signaling pathways, POXs are also involved in polymerization of suberin and lignin, important compounds of passive plant defense barriers [42] in a cross-talk with signaling pathways of jasmonic acid [47]. 

The apoplastic oxidative burst occurs as a result of ROS-producing activities of several groups of enzymes including cell wall peroxidases, plasma membrane NADPH oxidases, amine oxidases, lipoxygenases, oxalate oxidases and quinone reductases [48,49]. Lower apoplastic pH compared to the cytosolic pH has a crucial control effect on redox properties of protein cysteine thiols and overall redox conditions. Another important feature of the apoplastic space is decreased antioxidant capacity as a consequence of low abundance of low-molecular weight antioxidants such as glutathione and ascorbate. Decreased antioxidant capacity in the apoplast is critical for activation of ROS signaling pathways. Thus, the apoplastic oxidative burst results in enhanced apoplastic oxidation state, while cytoplasm stays reduced. Redox gradients formed across the cellular membrane allow for differential regulation of proteins on cell surface such as receptors important for stress signal perception [50]. Oxalate oxidase and amine oxidases belong to ROS sources found in the apoplast. Oxalate oxidase (EC 1.2.3.4) catalyzes oxidation of oxalate to H_2_O_2_ and CO_2_, and its gene expression is strongly induced by increased levels of salts and salicylic acid [51]. Copper-containing amine oxidases (CuAOs, EC 1.4.3.6) together with polyamine oxidases (PAOs, EC 1.5.3.3) are abundant plant proteins involved in the catabolism of polyamines, important plant growth regulators [52,53]. CuAOs are homodimer proteins contain copper ion and 2,4,5-trihydroxyphenylalanine quinone cofactor in both subunits. PAOs are monomeric and require flavin adenine nucleotide for their enzymatic activity. Enzymes of these classes are responsible for di- and polyamine catabolism, catalyzing oxidative deamination of their substrates, producing appropriate aldehydes, ammonia and H_2_O_2_ [54]. Produced apoplastic H_2_O_2_ is then used in modifications of the cell wall under physiological or stress conditions [55] or can function as a signal molecule, e.g., in the opening of Ca^2+^ channels [56,57].

Germins, a group of proteins with oxalate oxidases activity, represents another contribution to apoplastic ROS levels. Germins are apoplastic proteins with an oligomeric structure appearing at the onset of germination in plants, and are involved in plant development and defense responses to pathogen attack [58]. Oxalate oxidases contain manganese in their structure and are responsible for oxidation of oxalate. Moreover, plants with higher oxalate oxidase activity were found more resistant to plant pathogens [59]. Apart from germins, the germin protein family includes germin-like proteins (GLPs) which lack the oxalate oxidase activity but were shown to possess superoxide dismutase (SOD) activity [60].

NADPH oxidases (termed NOXs in animals) are family of proteins catalyzing production of superoxide in cytoplasmic membrane, termed as respiratory burst oxidase homologues (Rbohs) in plants [61,62]. Rbohs are considered the major ROS sources within the oxidative burst orchestrated in plant responses to pathogen attacks [63]. Within molecular events following the recognition of microbe- and damage-associated molecular patterns (MAMPs and DAMPs, respectively), the ROS is often observed as a biphasic ROS increase [64]. Rbohs were believed to specifically localize in plasma membrane [62]; however, recently, NtRbohD was observed localized in lipid rafts, membrane micro-domains in tobacco cells that can be coupled to other membrane components. It seems that different sub-membrane distribution of Rbohs is critical for their functions [65,66]. 

Protein structures of plant NADPH oxidases contain six conserved transmembrane helixes, where the 3rd and 5th helix bind two hem groups through four histidine residues. The hem groups are essential for transfer of electrons across the membrane to the molecular oxygen as the acceptor in the extracellular space [67]. N-terminal Ca^2+^ binding motifs and cytosolic FAD- and NADPH-binding domains in C-terminal regions are another typical structural feature of plant Rbohs. Plant Rbohs share a high similarity in their amino acid sequences of regions spanning membrane C-terminal cytosolic regions, whereas N-terminal cytosolic regions of NADPH oxidases show higher variability. NADPH oxidases require Ca^2+^ binding and phosphorylation to become fully activated; however, the exact activation mechanism has not been yet deciphered. These two activation events were suggested to occur synergistically [68], whereas other studies indicated the phosphorylation occurs as the first step followed by Ca^2+^ binding [69]. Both Rbohs activity and localization are controlled through phosphorylation by protein kinases from several families, including calcium-dependent and calmodulin-like domain protein kinases (CDPKs) [66,70]. The product of NADPH oxidase catalyzed reaction, superoxide anion radical is a membrane impermeable species due to its negative charge; however, under decreased pH conditions the superoxide is protonated and can pass throw the membranes. In this way, the pH status can affect compartmentalization of created product: in plants, where the physiological range is found around pH 5, approximately 16% of superoxide produced by Rbohs is transformed to a membrane permeable hydroperoxyl form [62].

Superoxide dismutases (SODs) constitute the first line of ROS catabolism with a crucial function in all compartments where superoxide radicals are produced [51]. Specifically in the apoplast, Cu-Zn SOD were reported to have a physiologically role in the lignification process in the vascular tissue of spinach hypocotyls [71]. It is noteworthy that concentrations of the key redox regulators glutathione (GSH) in the apoplast are orders of magnitude lower compared to the cytosol, where GSH is found in the milimolar range [72]. Furthermore, most apoplastic ascorbic acid (AA) in physiological conditions is oxidized, while over 90% of total leaf tissue AA is found in the reduced form [73], and oxidized AA derivatives, such as oxalic acid, were suggested to function as signaling molecules [74]. On the other hand, AA has been suggested to be the most important antioxidant for the detoxification of ROS in the apoplast in stress conditions [75]. Therefore, it seems that the low abundance of low molecular antioxidants in the apoplast might extend the half-life of H_2_O_2_, and thus enable H_2_O_2_ diffusion and propagation of ROS signals within and from the apoplast compartment.

### 2.2. Cytosol

The cytosol has not been regarded as an important compartment in plant ROS production. Cytoplasmic NADPH is a central component of redox maintenance in the cytosol as it controls the thiol/disulfide status and participates in the production of reducing substrates for antioxidant enzymes involved in ROS catabolism. On the other hand, NADPH also supplies electrons to plasma membrane NADPH oxidases as ROS producing enzymes [76]. Xanthine oxidase/dehydrogenase (XO; EC 1.2.1.37) and aldehyde oxidase (AO; EC 1.2.3.1) are molybdenum- and flavin-containing enzymes, which play important roles in plant purine metabolism and phytohormone biosynthesis, respectively. Both XO and AO contribute to increased ROS production in multiple animal pathologies and were suggested as potential sources of ROS also in plants. Interestingly, tomato and *Arabidopsis thaliana* XO was shown to produce superoxide but not H_2_O_2_ in vitro, whereas its animal counterpart can produce both superoxide and H_2_O_2_ [77]. Superoxide-generating activity was lost in *A. thaliana* Atxdh1 T-DNA insertion mutant and RNA interference lines, providing molecular evidence that plant XO generates superoxide. Unlike XO, AO produces only H_2_O_2_, which was surprisingly found not sensitive to inhibition by diphenyl iodonium, a known inhibitor of flavin-containing enzymes. Application of abscisic acid or water-stressed induced ROS production and AO and XO upregulation both in leaves and roots, indicating that plant AO and XO might be significant sources for ROS accumulation under stress condition [77].

Recently, AO isoforms AO2 and AO3 and their superoxide-producing activity were observed in *Nicotiana benthamiana* plants inoculated with the tomato bushy stunt virus, suggesting involvement of plant AO in defense mechanisms against viral infection [78]. It seems that another specific AO isoforms AsAO4 has a critical function in delaying senescence in siliques by catalyzing aldehyde detoxification under stress condition [79]. AAO4 enzyme oxidizes an array of aromatic and aliphatic aldehydes and generates superoxide and H_2_O_2_ in an aldehyde-dependent manner. Interestingly, XDH1 can play spatially specific dual and opposing roles in ROS metabolism in *A. thaliana* defense responses to powdery mildew [80]. In leaf epidermal cells, XDH1 functions as an oxidase, together with the NADPH oxidases RbohD and RbohF, to generate superoxide. Superoxide dismutation to H_2_O_2_ and subsequent accumulation of H_2_O_2_ in the fungal haustorial structures within infected epidermal cells contributes to constrain the haustorium, thereby contributing to powdery mildew resistance. In contrast, leaf XDH1 functions in the xanthine dehydrogenase mode producing uric acid in local and systemic tissues to scavenge H_2_O_2_ from stressed chloroplasts, thus contributing to plant protection from biotic stress-induced oxidative damage. 

Besides its role in ROS production, the cytosol is supposed to play a key role in redox signal integration. ROS signaling from the apoplast and organelles needs to pass through the cytoplasm to achieve its effects and modulate gene expression in the cell nucleus [81]. The signal can be transmitted directly from H_2_O_2_ or via redox sensitive proteins to the MAPK cascade and to redox-dependent transcription factors such as heat stress transcription factors [82]. A detailed discussion of ROS signaling mechanisms in plant cells is not within the scope of this paper; nevertheless, multiple aspects of plant ROS signaling have been recently covered elsewhere [9,11,12,29,36,61,83].

### 2.3. Mitochondria

Mitochondrial respiration, encompassed by a directed electron flow from reduced organic substrates to the molecular oxygen through components of respiratory chain in the inner mitochondrial membrane, is inheritable associated with ROS generation [84]. Therefore, superoxide production occurs during normal operation of the respiratory chain, but its rate is highly increased in conditions of decelerated respiratory rates, e.g., by respiratory chain inhibition or limited ADP availability, resulting in a highly reduced state of mitochondrial electron transport chain. Superoxide disproportionation to H_2_O_2_ and O_2_ is strongly accelerated by superoxide dismutase present in mitochondrial matrix [85].

In animals, respiratory complexes I and III are considered main sources of mitochondrial ROS, namely of superoxide anion radical, formed by a partial reduction of the molecular oxygen by highly reactive redox intermediates [86]. In plants, succinate-dependent H_2_O_2_ production in mitochondria was reported to be faster than pyruvate/malate-dependent H_2_O_2_ production, indicating a larger role for Complex II compared to Complex I [87]. Moreover, the ubiquinone pool might serve as another site of ROS production in plant mitochondria [88]. Using a *A. thaliana* mutant in the Complex II subunit succinate dehydrogenase 1-1, Complex II was shown to contribute to localized mitochondrial ROS production that regulates plant stress and defense responses [89].

ROS originating from plant mitochondria are known to impact both mitochondrial respiratory and cellular functions, including a variety of signaling cascades in the cell, comprising retrograde signaling, plant hormone action, programmed cell death and defense against pathogens [90]. Observed mitochondrial steady-state levels of H_2_O_2_ are two orders of magnitude lowers compared to peroxisomes and chloroplasts. However, in specific plant tissue or developmental stages, major production and accumulation of mitochondrial ROS can occur. During seed imbibition and hydration, high rates of mitochondrial respiration contribute as the major source of superoxide, which need to be properly balanced by the antioxidant machinery to avoid oxidative damage of biomolecules potentially resulting in defective seed germination [91]. Superoxide production in plant mitochondria can be minimized by several pathways that enable bypassing the electron transport chain. Proton leak across the membrane is facilitated by uncoupling proteins, whereas alternative oxidase (AOX) bypasses proton pumping on Complex III and IV [92]. Interestingly, overexpression of the *A. thaliana* uncoupling protein 1 in tobacco reduced ROS generation, induced the antioxidant system and increased resistance to diverse abiotic stress conditions [93,94].

### 2.4. Chloroplasts

Within photosynthetically active cells, chloroplasts act as another site of the major ROS production, connected with multiple redox reactions and electron transport chains localized in thylakoid membranes. Singlet oxygen (^1^O_2_) is a unique ROS species produced constitutively in plant leaves in light. Chlorophylls pigments in the antenna system and in the reaction center of photosystem II are primary sources of ^1^O_2_ in plants, which is formed in a reaction between the molecular oxygen and chlorophyll molecules in the triplet state [17,95,96]. The highly reactive nature of ^1^O_2_ is characterized by its half-life of about 3 μs in water solutions, but about 100 μs in hydrophobic environment of lipid membranes. A small part of singlet oxygen is able to diffuse to distances of several nanometers and it has been proposed that ^1^O_2_ can be involved in the signaling of programmed cell death or light acclimation processes [18]. Production of ^1^O_2_ is enhanced under light stress conditions, and lipoxygenase activity can provide an additional source of ^1^O_2_ [19]. Singlet oxygen can be detoxified namely by quenching by small lipophilic antioxidants like tocopherol or β-carotene. Carotenoids are capable of effective ^1^O_2_ scavenging resulting in formation of excited triplet states which return efficiently to the ground state through a heat release mechanism. Among carotenoids occurring in plant cells, β-carotene is functionally the most important [97]. Plastoquinone and D1 protein of photosystem II are known as other potential ^1^O_2_ scavengers [21]. Recently, a role of ^1^O_2_ produced by the chloroplast lipoxygenase was reported in leaf wounding using *A. thaliana lox2* mutants where ^1^O_2_ produced in wounded plants is involved in lipid or protein oxidation which can act as signaling mechanisms [20].

Chloroplast superoxide anion radical is known to occur both in photosystem I and II. In photosystem I, superoxide is generated by the Mehler reaction during low NADP^+^ concentrations and in presence of iron-sulfur proteins, reduced thioredoxin and ferredoxin [98]. Electron leakage to molecular oxygen on the acceptor side of photosystem II produces superoxide anion radical which after dismutation to H_2_O_2_ is reduced to hydroxyl radical by the non-hem iron. H_2_O_2_ formation by incomplete water oxidation occurs at the donor side of photosystem II, where it can be reduced to hydroxyl radical by manganese [99]. Another sources of chloroplastic ROS generation represent the oxygenase activity of ribulose-1,5-bisphosphate carboxylase/oxygenase (RuBisCO, EC 4.1.1.39) producing phosphoglycerate and phosphoglycolate that are metabolized by ongoing reactions in peroxisomes where H_2_O_2_ is formed [100]. 

Chloroplastic H_2_O_2_ is metabolized within glutathione-ascorbate cycle, called Foyer-Asada-Halliwell pathway, important cycle within the antioxidative mechanisms of plant cells [101,102,103]. Ascorbate peroxidase (APX, EC 1.11.1.11) scavenges H_2_O_2_ using ascorbate as an electron donor to form monodehydroascorbate (MDHA) as a first step of the pathway. Four APX isoforms are found within plant cells: thylakoid tAPX, chloroplast stromatal soluble sAPX, cytosol cAPX and glyoxysome membrane form mAPX. Then monodehydroascorbate reductase (MDHAR, EC 1.6.5.4) transforms MDHA to dehydroascorbate (DHA), which utilizes NADPH produced during photosynthesis to regenerate ascorbate. MDHAR also catalyzes the elimination of H_2_O_2_ from mitochondria and peroxisomes [104]. Ascorbate regeneration from DHA is mediated by dehydroascorbate reductase (DHAR, EC 1.8.5.1). Oxidized thiol groups of DHAR are then returned to a reduced state by reaction with reduced glutathione, which is in turn regenerated from the oxidized glutathione by glutathione reductase (GR, EC 1.6.4.2), enzyme with a critical role in redox pathways and essential for maintaining reduced pool of GSH [105]. Its localization is predominantly chloroplastic; however, minor GR activity is also found in cytosol and mitochondria [106].

### 2.5. Peroxisomes and Glyoxysomes

Plant peroxisomes accommodate multiple key metabolic pathways such as β-oxidation of fatty acids, photorespiration, ureides metabolism and xenobiotic detoxifications. Considerable amounts of ROS are produced in peroxisomal reaction pathways, peroxisomes are therefore included among major intracellular sources of plant ROS [107]. A short electron transport chain comprising cytochrome b and NADH-dependent ferric-chelate is located in the peroxisomal membrane (EC 1.16.1.7). Superoxide is formed on cytochrome b within the peroxisomal respiration and then released into the cytosol. Xanthine oxidase (XO, EC 1.17.3.2) contributes the peroxisomal ROS pool with superoxide formed during oxidation of xanthine or hypoxanthine to uric acid [108]. During photorespiration, H_2_O_2_ is produced in chloroplasts as a result of RuBisCO oxygenase activity followed by reaction of glycolate oxidase in peroxisomes [100]. Cu, Zn-SOD catalyzes dismutation of superoxide produced by matrix XO as another source of peroxisomal H_2_O_2_, which can be also produced by activities of several flavin oxidases. Furthermore, enzymes participating in ROS catabolism such as monodehydroascorbate reductase mediating superoxide detoxification are also present in peroxisomes [109].

Catalases (CAT, EC 1.11.1.6) are tetrameric hemoproteins catalyzing H_2_O_2_ decomposition to water and oxygen. CAT are primarily active in peroxisomes and glyoxysomes, the sites of high H_2_O_2_ generation and turnover [101]. According their catalytic mechanisms, CAT enzymes can be categorized into two groups: monofunctional with dismutating activity and bifunctional with dismutating/peroxidatic activities. By the current enzyme nomenclature, plant CAT isoforms comprise class I present in leaves and participating in H_2_O_2_ scavenging during photorespiration, class II are in vascular bundles, and CAT class III as key enzymes of H_2_O_2_ removal in the glyoxylate cycle [110].

Glyoxysomes were considered for long as a rather specialized organelle among plant microbodies derived from a single ancestral peroxisome, are important for specific plant aerobic pathways, especially glyoxylate cycle and β-oxidation of fatty acids [111]. However, recent studies accentuate unifying features of how these dynamic organelles contribute to energy metabolism, development and responses to environmental challenges. On the functional level, the major part of peroxisomal and glyoxysomal proteins are related to fatty acid oxidation. Analysis of triglyceride metabolism in *A. thaliana* seedlings showed that only two enzymes, isocitrate lyase and malate synthase, distinguish glyoxysomes from other peroxisomes [112]. Importantly, glyoxysomes as well as all peroxisome-like organelles share a number of characteristic enzymes related to ROS metabolism [107,113,114]. Nevertheless, glyoxysomes represent essential compartments of plant cells during seed germination [111,115]. Glyoxysomes are involved in mobilization of storage lipids in germinating seeds and represent an important source of both superoxide and H_2_O_2_ resulting from the β-oxidation of fatty acids and the activity of enzymes such as glycolate and urate oxidases [116,117]. Similarly to peroxisomes, glyoxysomes also contain catalase and H_2_O_2_-forming peroxidase.

### 2.6. Endoplasmic Reticulum

The endoplasmic reticulum (ER) is a dynamic organelle that fulfills many cellular functions including calcium storage, protein production and lipid metabolism. Metabolic pathways of protein biosynthesis, folding and post-translational modifications such as glycosylation, disulfide bond formation and chaperone-mediated protein folding processes are harbored in ER [118]. ER is also an essential cell compartment where oxidation, hydroxylation and deamination of cellular components or xenobiotics occur [119]. ROS generated in ER and moving the cytoplasm are partly produced in the electron transport chain located in membranes associated with ER; moreover, H_2_O_2_ is produced in ER for proper oxidative protein folding [120]. Stress-triggered H_2_O_2_ production can signal outside ER to induce components of antioxidant defense and balance the redox status of the cell [121,122]. In *A. thaliana*, ROS produced within the ER stress response pathway are involved the water stress-induced programed cell death [123]. Superoxide radical is a by-product of oxidation and hydroxylation processes involving cytochrome P450 and cytochrome P540 reductase in a presence of reduced NADPH [124]. 

## 3. Conclusions

In recent years, we have witnessed a substantial progress in our understanding of ROS-dependent redox signaling in plants involved in plant growth and developmental and responses to abiotic and biotic stress stimuli. It has become increasingly evident that cellular ROS signaling pathways are clearly confined in a spatio-temporal manner, similarly to that observed for other second messengers, and that ROS redox signaling is tightly connected to cellular compartmentalization, which allows organelle-specific signaling responses. Roles of low molecular antioxidants and ROS-catabolizing enzymes as important regulators of ROS levels in cellular compartments have been uncovered.

Within the plant ROS landscape, a simultaneous operation of both highly conserved and highly specific mechanisms of ROS production and degradation has been recognized in different plant cell organelles. It has become evident that light-driven reactions in thylakoid membranes can be considered the major ROS source in photosynthetically active tissues, whereas mitochondrial ROS production is of major importance in the dark or in non-green tissues. Furthermore, ROS production in the apoplastic space was uncovered to play a crucial role in plant interactions with external environment, including microbial organisms or root growth and development. However, some well-accepted concepts are continuously questioned by advances in the plant ROS field; e.g., singlet oxygen, generally thought to be produced in photosynthetic center, was reported to be involved in osmotic stress-induced cell death in *A. thaliana* roots, suggesting a novel light-independent mechanism of its generation [125].

Recent advances in the development and application of redox-sensitive probes enable specific measurements of individual ROS in different cellular compartments of intact plant cells. The application of these probes permits to analyze simultaneously spatio-temporal ROS modulations and functional associations between ROS metabolism and signaling and organelle functions. It is noteworthy that the current knowledge on the quantitative aspects of ROS metabolism in plant cells in vivo is still quite limited. This is mainly associated with current protocols used to detect ROS which are in many case not suitable for accurate and reliable ROS quantification [9,126]. Recent development of quantitative ROS transcription-based bioreporters seems a promising new strategy to achieve quantitative cellular mapping of ROS changes in plant responses to stress stimuli [127].

Moreover, actual advances of omics technologies provide new strategies to study in detail the site-specific functions of different components of ROS metabolism in plant cell signaling. With advances from single-cell to single-organelle transcriptomics, proteomics and metabolomics, specific roles of individual metabolites and enzymes in the regulation of ROS metabolism can be achieved. The recently introduced “ramanomics” approach might provide an efficient tool of non-invasive quantitative profiling of cellular compartments and monitoring of molecular interactions and transformations in live cells and their subcellular structures. Advancing field of the high-resolution microscopy combined with mentioned genetically-encoded redox sensors proteins is expected to provide new insights into the compartment-specific landscapes of short-lived reactive oxygen species in plant cell compartments, and to further elucidate their role in plant physiology and stress responses.

## Figures and Tables

**Figure 1 antioxidants-08-00105-f001:**
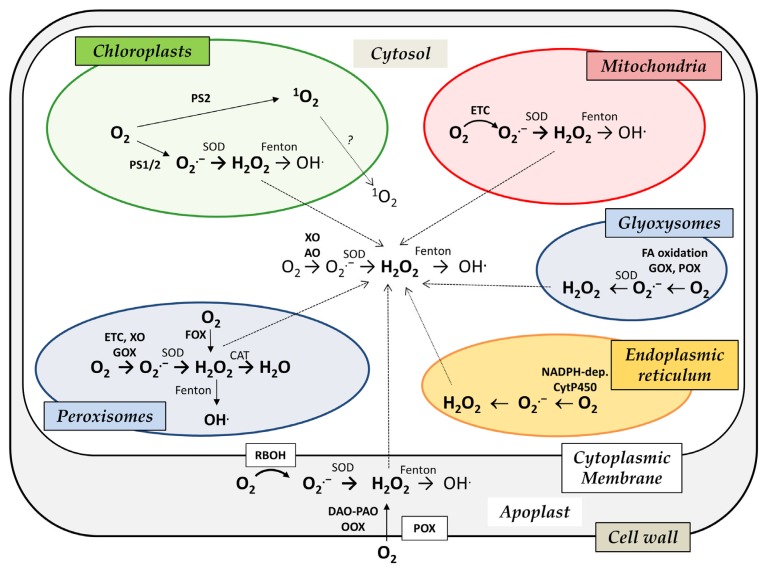
Schematic overview of reactive oxygen species (ROS) sources in plant cell compartments. AO, aldehyde oxidase; CAT, catalase; DAO, diamine oxidase; GOX, glycolate oxidase; ETC, electron transport chain; FOX, flavin oxidases; OOX, oxalate oxidase; PAO, polyamine oxidase; POX, peroxidases; PS, photosystem; RBOH, NADPH oxidase; SOD, superoxide dismutase; XO, xanthine oxidase/dehydrogenase.

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
