# Peer review of "On the Origin and Fate of Reactive Oxygen Species in Plant Cell Compartments"

_antioxidants, 2019, doi:10.3390/antiox8040105_

Round 1

Reviewer 1 Report

Several review papers about plant ROS were published recently, including:  Reactive Oxygen Species in Plant Signaling (2018) (doi: 10.1242/dev.164376), Reactive oxygen species in plant development (2018) (doi: 10.1242/dev.164376). In this review, the topic and content focus on the production, functions and signaling pathways of ROS based on the different cellular compartments should be of interest to many readers, but there is no big update on the published paper of the similar topic: especially in Reactive Oxygen Species in Plant Signaling (2018) (doi: 10.1242/dev.164376).  In this review, most of the references used were published before 2017. Only 4 references are after 2017, including three review papers. As a review paper, the authors should demonstrate the recent advances.

Suggest the authors change the title of section 2: Subcellular localization of ROS production in plant cells. The authors introduced the production, functions and signaling pathways of ROS based on the different cellular compartments, not only the Subcellular localization.

Suggest the authors introduce more recent advances in the section 2.

Suggest the authors add the content about the crosstalk of different ROS signalings.

Line 107: ROS termed the oxidative burst

Line 166: MAMPs/DAMPs (full name?)

Line 223-230: H2O2

line 264: mitochondrial ROS

Line 339: H2O2

Author Response

Antioxidants-467419

Authors: Martina Janků , Lenka Luhová, Marek Petřivalský

Title: On the origin and fate of reactive oxygen species in plant cell compartments

Reviewer 1

Several review papers about plant ROS were published recently, including:  Reactive Oxygen Species in Plant Signaling (2018) (doi: 10.1242/dev.164376), Reactive oxygen species in plant development (2018) (doi: 10.1242/dev.164376). In this review, the topic and content focus on the production, functions and signaling pathways of ROS based on the different cellular compartments should be of interest to many readers, but there is no big update on the published paper of the similar topic: especially in Reactive Oxygen Species in Plant Signaling (2018) (doi: 10.1242/dev.164376).  In this review, most of the references used were published before 2017. Only 4 references are after 2017, including three review papers. As a review paper, the authors should demonstrate the recent advances.

Reply:

We have replaced selected citations with more recently published papers:

9.Mhamdi, A.; Van Breusegem, F. Reactive oxygen species in plant development. Development 2018, 145, dev164376.

10.Choudhury, F.K.; Rivero, R.M.; Blumwald, E.; Mittler, R. Reactive oxygen species, abiotic stress and stress combination. Plant J. 2017, 90, 856-867.

13.Eckardt, N.A. The Plant Cell Reviews Plant Immunity: Receptor-Like Kinases, ROS-RLK Crosstalk, Quantitative Resistance, and the Growth/Defense Trade-Off. Plant Cell 2017, 29, 601-602.

19.Dogra, V.; Rochaix, J.D.; Kim, C. Singlet oxygen‐triggered chloroplast‐to‐nucleus retrograde signalling pathways: An emerging perspective. Plant Cell Environ. 2018, 41, 1727– 1738

25.Tamma, G.; Valenti, G.; Grossini, E.; Donnini, S.; Marino, A.; Marinelli, R.A., Calamita, G. Aquaporin Membrane Channels in Oxidative Stress, Cell Signaling, and Aging: Recent Advances and Research Trends. Oxid. Med. Cell.Longev. 2018, 1501847.

26.Nordzieke, D.E.; Medraño-Fernandez, I. The Plasma Membrane: A Platform for Intra- and Intercellular Redox Signaling. Antioxidants 2018, 7, pii: E168.

28.Jacoby, R. P.; Millar, A. H.; Taylor, N. L. Mitochondrial Biochemistry: Stress Responses and Roles in Stress Alleviation. In Annual Plant Reviews online, J. A. Roberts (Ed.), 2018.

30.Srivastava, A.; Redij, T.; Sharma, B.; Suprasanna, P. Interaction between Hormone and Redox Signaling in Plants: Divergent Pathways and Convergent Roles. In Mechanism of Plant Hormone Signaling under Stress, G. K. Pandey (Ed.), 2017.

35.Zipfel, C. Plant pattern-recognition receptors. Trends Immunol. 2014, 35, 345-351.

37.Schmidt, R.; Kunkowska, A. B.; Schippers, J. H. Role of Reactive Oxygen Species during Cell Expansion in Leaves. Plant Physiol. 2016, 172, 2098–2106.

39.Lazzarotto, F.; Turchetto-Zolet, A.C.; Margis-Pinheiro; M. Revisiting the Non-Animal Peroxidase Superfamily.Trends Plant. Sci. 2015, 20, 807-81383.Cezary, W.; Carmody, M.; Kangasjärvi, J. Reactive Oxygen Species in Plant Signaling. Annu. Rev. Plant Biol. 2018, 69, 209-236.

42.Lüthje, S.; Martinez-Cortes, T. Membrane-Bound Class III Peroxidases: Unexpected Enzymes with Exciting Functions. Int. J. Mol. Sci. 2018, 19, 2876.

48.Survila, M.; Davidsson, P.R.; Pennanen, V.; Kariola, T.; Broberg, M.; Sipari, N.; Heino, P.; Palva, E.T. Peroxidase-Generated Apoplastic ROS Impair Cuticle Integrity and Contribute to DAMP-Elicited Defenses. Front. Plant Sci. 2016, 23, 1945.

54.Chen, D.; Shao, Q.; Yin, L.; Younis, A.; Zheng, B. Polyamine Function in Plants: Metabolism, Regulation on Development, and Roles in Abiotic Stress Responses. Front. Plant Sci. 2019, 9, 1945.

55.Tavladoraki, P.; Cona, A.; Angelini, R. Copper-Containing Amine Oxidases and FAD-Dependent Polyamine Oxidases Are Key Players in Plant Tissue Differentiation and Organ Development. Front. Plant Sci. 2016, 7, 824.

59.Zhang, X. Y.; Nie, Z. H.; Wang, W. J.; Leung, D. W.; Xu, D. G.; Chen, B. L.; Liu, E. E. Relationship between disease resistance and rice oxalate oxidases in transgenic rice. PloS One 2013, 8, e78348

125.Chen, T.; Fluhr, R. Singlet Oxygen Plays an Essential Role in the Root’s Response to Osmotic Stress. Plant Physiol. 2018, 177, 1717-1727.

126.Noctor, G.; Foyer, C. H. Update on redox compartmentation intracellular redox compartmentation and ROS-related communication in regulation and signaling. Plant Physiol. 2017, 171, 1581-1592.

127.Lim, S.D.; Kim, S.H.; Gilroy, S.; Cushman, J.C.; Choi, W.G. Quantitative ROS bioreporters: A robust toolkit for studying biological roles of ROS in response to abiotic and biotic stresses. Physiol Plant. 2019, 165, 356-368.

Suggest the authors change the title of section 2: Subcellular localization of ROS production in plant cells. The authors introduced the production, functions and signaling pathways of ROS based on the different cellular compartments, not only the Subcellular localization.

Reply: As suggested, we have changed the title of section 2 to “Subcellular localization and functions of ROS production in plant cells”.

Suggest the authors introduce more recent advances in the section 2.

Reply: As mentioned above, we have included new parts citing more recent papers throughout the manuscript when appropriate.

Suggest the authors add the content about the crosstalk of different ROS signalings.

Reply: As suggested by reviewer 2, we have rewritten the corresponding manuscript part to reflect more accurately the main focus of the manuscript in relation to ROS signalling, including citations of recent reviews on plant ROS signalling.

Line 107: ROS termed the oxidative burst

Line 166: MAMPs/DAMPs (full name?)

Line 223-230: H2O2

line 264: mitochondrial ROS

Line 339: H2O2

Reply: We have performed all suggested corrections.

Reviewer 2 Report

In the present manuscript, the authors review plant ROS homeostasis in plant cells.  The chemical focus on the mechanism of production in the different organelles clearly distinguishes this manuscript from many other excellent recent reviews on the topic of ROS signalling in plants and clearly is a strength of this manuscript.
To further support this focus, I would suggest to rearrange the chapters on the organelles. For each chapter, use the same structure: production of ROS, degradation of ROS, physiological implications and maybe use sub-headings to make this clear. Subsequently, in the conclusions, it would be great to have some comparison between the organelles: which enzymes are used in common, where are the differences (e.g. light induced ROS in the chloroplast clearly stands out).
Furthermore, it would be great to have some numbers. How much does the chloroplast/ mitochondria… contribute to overall ROS? This clearly changes depending on the environmental conditions, but summarizing the current knowledge could help building mathematical models on ROS in plant cells.

Minor points:

Chapter 2.1 (line 100-115): In this introductory paragraph, I am missing the role of apoplastic ROS in plant development. For example, they are essential for cell wall remodelling during root elongation.

Line 107: please correct the typo: ROSS à ROS

Line 201 ff: Please use Arabidopsis thaliana and A. thaliana instead of Arabidopsis to be specific.

Line 232-240: ROS signalling in plants is an active field. There is much more information available than presented here. It would be good to state that the details of signalling are not the scope of this review and refer to current literature on ROS signalling.

Author Response

Antioxidants-467419

Authors: Martina Janků , Lenka Luhová, Marek Petřivalský

Title: On the origin and fate of reactive oxygen species in plant cell compartments

Reviewer 2

In the present manuscript, the authors review plant ROS homeostasis in plant cells.  The chemical focus on the mechanism of production in the different organelles clearly distinguishes this manuscript from many other excellent recent reviews on the topic of ROS signalling in plants and clearly is a strength of this manuscript. To further support this focus, I would suggest to rearrange the chapters on the organelles. For each chapter, use the same structure: production of ROS, degradation of ROS, physiological implications and maybe use sub-headings to make this clear.

Reply: We have seriously considered this reviewer suggestion; however, we feel that in most chapters on the organelles we have followed in major the same structure as suggested. In some instances, we have considered more appropriate not to follow this so strictly, in order to illustrate more clearly the mechanistic connections among various ROS species produced in one organelle.

Subsequently, in the conclusions, it would be great to have some comparison between the organelles: which enzymes are used in common, where are the differences (e.g. light induced ROS in the chloroplast clearly stands out).

Reply: We appreciate this suggestion and we have rewritten the conclusion section to include an appropriate comparison of organelle-specific ROS production mechanisms.

Furthermore, it would be great to have some numbers. How much does the chloroplast/ mitochondria… contribute to overall ROS? This clearly changes depending on the environmental conditions, but summarizing the current knowledge could help building mathematical models on ROS in plant cells.

Reply: We appreciate this comment, but we consider that actually there are not data available enabling accurate estimations of contribution of individual ROS sources in vivo in plant cells, which would be suitable to build mathematical models. In this respect, we have introduced new sentences in the Conclusion part on the quantitative aspects of ROS metabolism in plants.

Minor points:

Chapter 2.1 (line 100-115): In this introductory paragraph, I am missing the role of apoplastic ROS in plant development. For example, they are essential for cell wall remodelling during root elongation.

Reply: The role of apoplastic ROS was already briefly mentioned in line 114-116, but as suggested, we have including additional information in this manuscript section on the role of apoplastic ROS.

Line 107: please correct the typo: ROSS à ROS

Reply: Corrected.

Line 201 ff: Please use Arabidopsis thaliana and A. thaliana instead of Arabidopsis to be specific.

Reply: As suggested, we have corrected this issue by using consistently the name of A. thaliana in the entire manuscript.

Line 232-240: ROS signalling in plants is an active field. There is much more information available than presented here. It would be good to state that the details of signalling are not the scope of this review and refer to current literature on ROS signalling.

Reply: As suggested, we have rewritten this manuscript part to reflect more accurately the main focus of the manuscript in relation to ROS signalling, including corresponding citations to recent reviews on plant ROS signalling.

Round 2

Reviewer 1 Report

agree

Reviewer 2 Report

Thank you very much for considering most of my comments. I am happy with the current version and will surely use it not only in scientific context, but also for teaching.

One Detail that I spotted: 

Line106 and line 109: “termed as the oxidative burst” is repeated in one sentence